# Combined self-supervised learning for ECG classification based on the classification token

Doga Gündüzalp
*Department of Computer Engineering*
*Technical University of Munich*
Munich, Germany
doga.gunduzalp@tum.de

Congyu Zou
*Department of Computer Engineering*
*Technical University of Munich*
Munich, Germany
congyu.zou@tum.de

*Abstract*—Electrocardiogram (ECG) is a crucial non-invasive method for measuring the electrical activities of the heart and detecting cardiovascular diseases. While deep learning approaches for cardiovascular disease classification have gained popularity, creating labeled data remains expensive. Contrastive and generative self-supervised learning methods, particularly with the vision transformer (ViT) neural network architecture, have been introduced as leading solutions to this challenge. Although the [CLS] token, which is one of the most important components of ViT, is frequently utilized for aggregate representation in supervised learning scenarios, such as diagnosis, its exploitation in self-supervised tasks has not been extensively explored. This study proposed a method to incorporate multiple pretraining tasks for better representation learning via utilizing the [CLS] token more effectively. Based on this method, we introduced two novel combined self-supervised learning frameworks for ECG analysis, namely MAE-MoCo and MAE-Nextclip. The MAE-MoCo framework combines generative and contrastive self-supervised learning by incorporating a masked autoencoder with a momentum encoder. On the other hand, MAE-Nextclip is a generative method that reconstructs not only masked patches but also Nextclip data with the assistance of the [CLS] token. We validated our methods on a joint database of China Physiological Signal Challenge in 2018, Physikalisch-Technische Bundesanstalt XL, and Chapman datasets. The fine-tuned models outperformed the state-of-the-art models in the Chapman dataset with macro-F1 of 0.96 and the area under the curve of receiver operating characteristic of 0.99. The outstanding performance on downstream tasks demonstrates the potential of combining pretraining tasks, especially generative and contrastive tasks, in the field of automatic ECG interpretation.

*Index Terms*—classification token, electrocardiogram, generative-contrastive learning, self-supervised learning, vision transformer

## I. Introduction

Electrocardiogram (ECG) is a non-invasive method that records activities by measuring the electrical signals of the heart [1]. Diseases related to the heart, also known as cardiovascular diseases, hold significant importance among causes of death [2]. ECG stands as one of the most effective and prevalent tools used in the diagnosis of cardiovascular diseases due to its ability to promptly generate results, non-invasiveness, and capability to measure the activities of the heart, the most vital organ. Due to probes attached to different points of the body, ECG signals consisting of 12 leads encapsulate the electrical activity of the heart and this is done for a certain period of time. Due to these attributes, ECG signals can be treated both as a time series and as a two-dimensional medical image.

ECG data has been identified as a compelling area of study within the field of machine learning (ML) research. ML models have gained significant value in recent years, showcasing impressive performances in various fields such as computer vision (CV) [3], natural language processing (NLP) [4], [5]. The successful operation of non-linear algorithms with ECG signals signifies the potential effectiveness of ML-based models in the field of ECG. The performance of traditional ML algorithms such as Gaussian Naive Bayes and random forest stands out in heart disease classification [6], while the impact of deep learning models in arrhythmia beat classification has also been demonstrated by numerous studies [7]. Especially from the CV perspective, convolutional neural network (CNN) based architectures such as ResNet [8], InceptionTime [9] and from the time series perspective, recurent neural network (RNN) based long short-term memory (LSTM) serve as highly important backbone candidates [10], [11].

Supervised learning in ECG analysis requires extensive labeled data, often necessitating significant time and expertise, particularly from cardiologists, to label such medical data accurately. Due to the widespread use of ECGs, large volumes of unlabeled data are available, yet labeling these for supervised learning is labor-intensive. In contrast, self-supervised learning leverages unlabeled data, enabling models to generate their own labels, thus addressing the limitations of supervised learning. Self-supervised learning is particularly beneficial when labeled data is scarce and may outperform the models trained solely on supervised data [12]. Self-supervised learning methods, categorized into generative, contrastive, and adversarial learning [13], produce pretrained encoders that serve as robust feature extractors for downstream tasks, highlighting the growing importance of self-supervised learning in ECG analysis.

In the field of ECG, there exist frameworks derived from the two primary branches of self-supervised learning, namely contrastive and generative learning perspectives, demonstrating exceptional performance. Within the contrastive learning perspective, it has been shown that simple contrastive methods, including end-to-end trained SimCLR [14] are effective.

Additionally, models like MoCo [15] and BYOL [16], which use gradient descent for generating query and key representations in one branch and momentum updates in another, also demonstrate significant performance. Although SimCLR, MoCo, and BYOL are effective contrastive self-supervised learning methods, none of them are designed specifically for ECG. One of the most effective contrastive learning methods designed specifically for ECG is ISL, as it is tailored for multivariate cardiac signals [17]. This advanced contrastive learning method combines intra-subject and inter-subject self-supervised techniques. The intra-subject approach examines temporal dependencies in ECG signals from the same patient, while the inter-subject approach computes contrastive loss by treating augmented signals from the same patient as positive pairs and signals from different patients as negative pairs.

From the generative learning perspective, the models presented in the ECG domain are primarily based on masked autoencoding (MAE) [18]. Models such as masked time autoencoder (MTAE) [19], which is pretrained to find the masked ECG patches in the time axis, masked lead autoencoder (MLAE) [19], which reconstructs masked patches along the lead axis, are among the state-of-the-art performing models utilizing masking in the ECG classification domain. In addition, MassMIB, another method based on masking technique, encompasses an encoder-decoder pair that encodes a signal masked in the time domain and reconstructs the masked tokens in time, alongside an additional encoder-decoder pair that performs the same task in the frequency domain. Lastly, Nextclip [20] reconstructs the next semi-cardiac cycle by removing ECG data after any given time point as a pretraining task.

Recent advancements in contrastive self-supervised learning for ECG data, such as SimCLR, MoCo, and BYOL, have been significant but the rise of transformer-based architectures coupled with methods like MAE has led to state-of-the-art performance in generative self-supervised learning methods. The backbone architecture of the encoder is crucial for performance, with vision transformer (ViT) outperforming ResNet [21] in many ECG-specific pretraining tasks [19]. A notable feature of ViT is the inclusion of the [CLS] token, which, as seen in BERT, is beneficial for aggregate representation. However, in ECG pretraining tasks, the [CLS] token does not leverage its characteristic of carrying aggregate representation effectively, suggesting the potential for more optimal approaches in pretraining.

This paper introduces a novel approach that combines pretext tasks in generative and contrastive learning perspective using the [CLS] token to enhance ECG signal representation. The primary objective is to create a robust pretraining task by incorporating the [CLS] token of ViT into a hybrid learning framework for ECG signal representation. In this context, two approaches are explored: adding a contrastive projection head to the [CLS] token output of a MAE framework using 1D-ViT called as **MAE-MoCo**, and incorporating a new Nextclip decoder to create a combined generative self-supervised learning framework named as **MAE-Nextclip**. MAE-MoCo aims to develop an encoder that targets both the reconstruction of the ECG signal and the differentiation of positive and negative pairs, while MAE-Nextclip reconstructs the masked patches as well as the ECG data in the next semi-cardiac cycle, outperforming current state-of-the-art techniques. In addition, the novel loss functions tailored to these combined pretraining tasks are introduced. The experimental results highlight the effectiveness of the proposed MAE-MoCo and MAE-Nextclip frameworks, demonstrating their potential to set new benchmarks in self-supervised learning methods for ECG signal analysis.

## II. METHOD

The proposed methodology includes two novel pretraining techniques, namely MAE-MoCo and MAE-Nextclip, which are later fine-tunable on specific datasets. Both models effectively utilize the [CLS] token of ViT in their pretraining tasks. The MAE-MoCo model merges generative and contrastive learning perspectives, whereas MAE-Nextclip combines two distinct generative learning tasks.

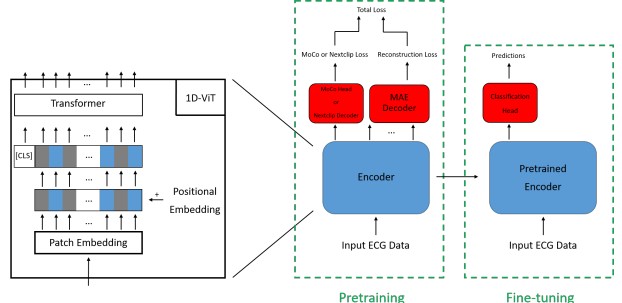

Fig. 1. Rough graphical illustration of an overall methodology for this paper: A novel approach to pretraining an ECG encoder. Blue patches represent unmasked, gray patches represent masked patches.

### A. Model Architectures

*1) MAE-MoCo:* The MAE-MoCo model consists of three main blocks as *encoder*, *decoder*, and *momentum encoder*.

The encoder is based on a 1D-ViT structure and comprises two parts for MAE and MoCo. The MAE part of the encoder takes the original ECG data without any data augmentation and first passes it through a *patch embedding* block. This block, which is a 1D CNN block with kernel size and stride equal to the patch size, divides the ECG data into patches of a specified size and maps each patch to the hidden size. Next, *positional embedding* is added to each patch to ensure that positional information is not lost. Although a fixed sine-cosine embedding is used for *positional embedding* in [18], this study opts for learnable *positional embedding* parameters since the encoder is used for both MAE and MoCo. Following this, a certain percentage of patches are masked using *random masking*. It should be noted that, only the unmasked patches are used afterwards. Finally, the masked encoded ECG signal is obtained by passing through six consecutive stacked *transformer* blocks and a *layer normalization* block.

The MoCo part of the encoder first processes the ECG input with a randomly selected data augmentation technique. It passes through the *patch embedding* and *positional embedding* layers without masking, in a similar manner with MAE part. Instead of masking, a learnable parameter called [CLS] token is added to represent the overall structure of the ECG signal. The signal then goes through the *transformer* block and *layer normalization* block. Finally, only the [CLS] token is extracted and passes through the Linear+Batch Normalization+ReLU block to obtain the encoded MoCo output. The block diagram of the encoder of the MAE-MoCo model is given in Fig. 2.

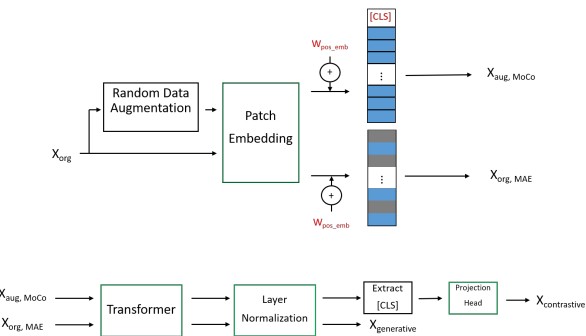

Fig. 2. The encoder block diagram of the MAE-MoCo model is depicted. In figure, $x_{org}$ represents the original ECG data, $x_{generative}$ represents the masked latent representation used for generative learning (MAE), and $x_{contrastive}$ represents the latent representation used for contrastive learning (MoCo). The parameters marked in red are learnable parameters, while the green blocks represent learnable blocks. Masked patches are represented in gray, while unmasked patches are represented in blue.

The purpose of the decoder network is to reconstruct the masked parts with the least error by replacing them with learnable parameters based on the encoded output for generative learning ($x_{generative}$) in the encoder. The decoder block takes the $x_{generative}$ and first passes it through a *patch embedding* block. This block is structurally different from the encoder's patch embedding block because the decoder does not perform patch separation; instead, patches are mapped to a different dimension using a linear layer. Subsequently, a learnable [mask] token is placed at the position of the masked patches resulting from random masking in the encoder, and *positional embedding* is added to each patch. Next, similar to the encoder, the decoder employs *transformer* and *layer normalization* blocks. Finally, another linear layer is used to match the output dimension with the original ECG, aiming to reconstruct the masked ECG patches with minimum error. The overall structure of the decoder is given in Fig. 3.

The momentum encoder in the MAE-MoCo framework uses the same architecture but ignores the output $x_{generative}$. Additionally, the data augmentation technique used is different from that in the encoder. Similar to MoCo, the parameters of the momentum encoder are trained with the momentum update of the encoder, not with the gradient descent. With each batch, the oldest batch in the queue is removed, and a new batch is added, following the principles of MoCo training.

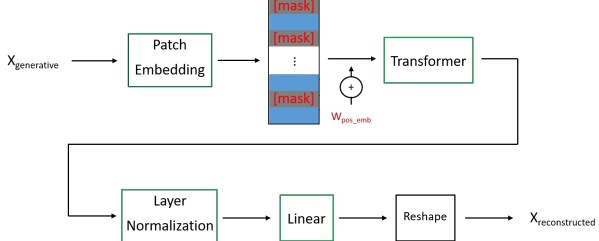

Fig. 3. The overall structure of the decoder. To reconstruct the masked patches, a learnable [mask] token is placed in their place.

A summarized working scheme of the MAE-MoCo framework is given in Fig. 4 and detailed hyperparameters of the encoder, decoder, and momentum encoder are described in Table I, II and III, respectively. The architectural hyperparameters of the momentum encoder are exactly the same as the encoder.

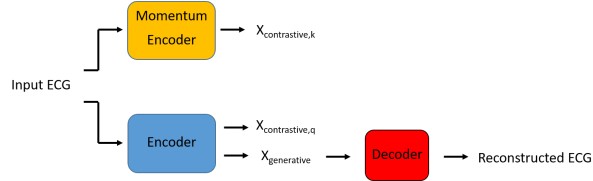

Fig. 4. The summarized working scheme of the MAE-MoCo framework. It should be noted that although the encoder and momentum encoder have the same architecture, they use different data augmentation techniques.

TABLE I
HYPERPARAMETERS OF THE ENCODER OF MAE-MOCO FRAMEWORK

| Encoder | Hyperparameters |
|---|---|
| Data augmentation techniques | Erase, time out, partial noise, drop and RRC |
| Type of patch embedding | 1D Convolution |
| Output channels of patch embedding | 512 |
| Kernel size of patch embedding | 25 |
| Stride of patch embedding | 25 |
| Size of positional embedding | (145, 512) |
| Size of [CLS] token | (1,512) |
| Type of transformer block | Attention and linear |
| Dimension of transformer | 512 |
| #Transformer blocks | 6 |
| #Attention head | 16 |
| MLP ratio of transformer | 2 |
| Activation of transformer | GELU |
| Output features of layer normalization | 512 |
| Type of the projection head | Linear and 1D BatchNorm |
| #Blocks in projection head | 3 |
| Output features of MLP in projection head | 128 |
| Output features of projection head | 128 |

*2) MAE-Nextclip:* The MAE-Nextclip framework consists of a shared encoder and two separate decoders for the tasks of MAE and Nextclip reconstruction, respectively. Architecturally and functionally, this model bears significant resemblance to the MAE-MoCo model.

The MAE component of the encoder is identical to that described in Section II-A1. For the Nextclip part, the original ECG data is first transformed into Nextclip data. In this

TABLE II
HYPERPARAMETERS OF THE DECODER OF MAE-MoCo FRAMEWORK

| Decoder | Hyperparameters |
|---|---|
| Type of patch embedding | Linear |
| Output channels of patch embedding | 256 |
| Size of `[mask]` token | (1,256) |
| Size of positional embedding | (144, 256) |
| Type of transformer block | Attention and linear |
| Dimension of transformer | 256 |
| #Transformer blocks | 1 |
| #Attention head | 8 |
| MLP ratio of transformer | 2 |
| Activation of transformer | GELU |
| Output features of layer normalization | 256 |
| Type of size matcher network | Linear |
| Output channels of size matcher network | 25*12 |

TABLE III
HYPERPARAMETERS OF THE MOMENTUM ENCODER OF MAE-MoCo FRAMEWORK

| Momentum Encoder | Hyperparameters |
|---|---|
| Momentum constant | 0.99 |
| Temperature of logits | 0.2 |
| Queue size | 65536 |

context, a random starting point is selected within a patch of the ECG data. A semi-cardiac cycle is then extracted from this randomly selected point and used as the ground truth data, and the ECG data from this point onward is set to zero for use in the rest of the encoder. The prepared ECG data for Nextclip is embedded with *patch embedding* and *positional encoding*. Additionally, a `[CLS]` token is appended for use in the Nextclip decoder. Finally, the resulting signal undergoes encoding through stacked *transformer* blocks and *layer normalization*. The operational schematic of MAE-Nextclip encoder is provided in Fig. 5.

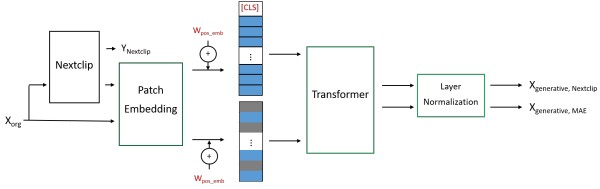

Fig. 5. The working scheme of the MAE-Nextclip encoder. $y_{Nextclip}$ represents the ground truth ECG cycle intended for reconstruction after the Nextclip decoder and $x_{generative,Nextclip}$ and $x_{generative,MAE}$ respectively denote the signals encoded for Nextclip and MAE.

The MAE decoder is identical to the one depicted in Fig. 3. However, the Nextclip decoder initially extracts the `[CLS]` information from the $x_{generative,Nextclip}$ signal and then proceeds with *patch embedding*. Afterward, without introducing a `[mask]` token, it sequentially traverses through *transformer*, *layer normalization*, and *linear* layer to reconstruct the Nextclip signal. A summary of the decoders used separately for MAE and Nextclip is provided in Fig. 6.

The overall block diagram of the MAE-Nextclip model is given in Fig. 7. Since the hyperparameters of the MAE-Nextclip model exhibit substantial similarities with those of

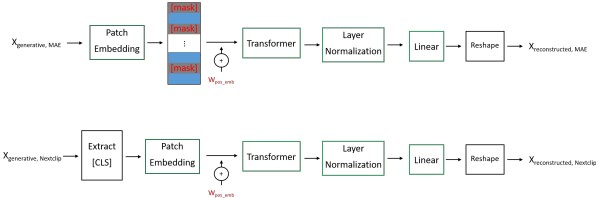

Fig. 6. The summary of the MAE-Nextclip decoders.

MAE-MoCo, it is more efficient to focus solely on the points of differentiation. Specifically, the hyperparameters of the MAE-Nextclip encoder, including *patch embedding*, *positional embedding*, `[CLS]` token, *transformer*, and *layer normalization*, uses those detailed in Table I. It should be noted that the encoder of MAE-Nextclip model does not employ data augmentation techniques and lacks the projection head presented in MoCo. The hyperparameters of the MAE decoder in MAE-Nextclip correspond exactly to those delineated in Table II, with each element in active use. The hyperparameters specific to the Nextclip decoder of MAE-Nextclip are provided in Table IV. Furthermore, the logic behind the selection of the 9 in the size matcher network is rooted in the attempt to predict a cardiac cycle of length 9 patches, starting from where the data is cropped during the creation of Nextclip data.

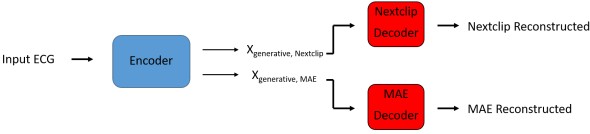

Fig. 7. The overall block diagram of the MAE-Nextclip model.

TABLE IV
HYPERPARAMETERS OF THE NEXTCLIP DECODER OF MAE-NEXTCLIP MODEL

| Nextclip Decoder | Hyperparameters |
|---|---|
| Type of patch embedding | Linear |
| Output channels of patch embedding | 256 |
| Size of positional embedding | (1, 256) |
| Type of transformer block | Attention and linear |
| Dimension of transformer | 256 |
| #Transformer blocks | 1 |
| #Attention head | 8 |
| MLP ratio of transformer | 2 |
| Activation of transformer | GELU |
| Output features of layer normalization | 256 |
| Type of size matcher network | Linear |
| Output channels of size matcher | 9*25*12 |

### B. Training Strategies

The pretraining and fine-tuning of the models utilized an Intel(R) Core(TM) i7-11800H @ 2.30GHz CPU, 16GB RAM, and an NVIDIA GeForce RTX 3060 GPU.

*1) Pretraining:* The pretraining of the models utilized a combination of three different datasets namely China

Physiological Signal Challenge in 2018 (CPSC2018) [22], Physikalisch-Technische Bundesanstalt XL (PTB-XL) [23], and Chapman [24]. Due to the high number of ECG samples and the diversity of data collected from various hospitals, the dataset exhibits significant diversity. Therefore, it is highly valuable for pretraining.

In order to enhance the stability of the outputs obtained from the models with the provided input data, the inputs are Z-score normalized. In other words, the mean of the input data is subtracted and divided by its standard deviation.

$$x_{normalized} = \frac{x_{org} - \sigma}{s} \qquad (1)$$

Due to the composition of the MAE-MoCo and MAE-Nextclip models from two different pretraining techniques, conventional loss functions cannot be directly utilized. Instead, a combination of these loss functions is employed. The MAE-MoCo model comprises a combination of generative and contrastive loss functions. The generative loss function calculates the MSE between the original data and the reconstructed data of masked patches. The generative loss is illustrated by (2) [19].

$$\mathcal{L}_{MAE} = \frac{1}{n} \sum_{i=1}^{n} (x_{org,i} - x_{reconstructed,i})^2 \qquad (2)$$

For the contrastive loss, the instance discrimination method is utilized. Thus, if the encoded keys and queries originate from the same ECG, they are considered as positive pairs; otherwise, they are treated as negative pairs. The contrastive loss is mathematically described in (5) [15], where $\ell_{pos}$ and $\ell_{neg}$ represent the positive and negative logits, respectively, $[]$ denotes concatenation operation, and $CE$ stands for cross-entropy loss function.

$$\ell_{pos} = x_{cont,q} * x_{cont,k} \qquad (3)$$
$$\ell_{neg} = x_{cont,q} * queue \qquad (4)$$
$$\mathcal{L}_{MoCo} = CE([\ell_{pos}, \ell_{neg}], labels) \qquad (5)$$

After the computation of the generative and contrastive losses, these two loss functions are combined for the MAE-MoCo model as depicted in (6).

$$\mathcal{L}_{MAE-MoCo} = \mathcal{L}_{MoCo} + \alpha \mathcal{L}_{MAE} \qquad (6)$$

Similarly, the MAE-Nextclip model employs (2), where the variable $x_{reconstructed,i}$ is substituted with the output $x_{reconstructed,MAE}$ of the MAE-Nextclip model. Since the MAE-Nextclip model is a combination of two different generative methods, it utilizes the Nextclip loss instead of the contrastive loss. As demonstrated in (7), the Nextclip loss is computed as the MSE between the reconstructed cardiac cycle and the ground truth. The overall MAE-Nextclip loss function is illustrated in (8).

$$\mathcal{L}_{Nextclip} = \frac{1}{n} \sum_{i=1}^{n} (y_{Nextclip} - x_{reconstructed,Nextclip})^2 \quad (7)$$

$$\mathcal{L}_{MAE-Nextclip} = \mathcal{L}_{Nextclip} + \alpha \mathcal{L}_{MAE} \qquad (8)$$

The $\alpha$ parameter in the loss function of both models is a hyperparameter and experiments are conducted with different $\alpha$ values. However, in the initial case, the $\alpha$ is set to 40 for MAE-MoCo and 2 for MAE-Nextclip. The reason for this is to bring both losses to the same scale. Additionally, training has been conducted for MAE-MoCo with $\alpha$ values of 20 and 10, while for MAE-Nextclip, configurations with $\alpha$ values of 1 and 0.5 have been employed.

Both models update their own weights to minimize their respective combined batch-wise losses during training. Warm-up epochs are applied by linearly increasing the learning rate for the initial specified number of epochs. Warm-up epochs serve to stabilize the training process and accelerate convergence [25]. Throughout the training process, the learning rate is reduced as epochs progress with the aid of learning rate schedulers. If there is no improvement in the validation loss after a certain number of epochs, the learning rate decreases by a specified rate. The detailed configuration of the models used for pretraining is provided in Table V.

TABLE V
PRETRAINING CONFIGURATION

| | |
|---|---|
| Epochs of MAE-MoCo | 710 |
| Epochs of MAE-Nextclip | 620 |
| Batch size | 64 |
| Optimizer | Adam |
| Learning rate | $2 * 10^{-4}$ |
| Warm-up epochs | 20 |
| Learning rate schedular: reduce rate | 0.9 |
| Learning rate schedular: number of bad epoch | 15 |

It should be noted that in addition to the hyperparameters and pretraining configuration parameters of models, there are certain fixed parameters used in models as well. The length of the ECG data is 3600, with 12 channels. The length of patches used by ViT is chosen as 25 and 75% of the patches are masked in MAE. Therefore, the models have 144 tokens, excluding the [CLS] token. The fixed parameters are listed in Table VI.

TABLE VI
FIXED PARAMETERS

| | |
|---|---|
| ECG signal length | 3600 |
| #Leads of ECG data | 12 |
| Patch size | 25 |
| Masking rate | 75% |
| #Tokens (without [CLS] token) | 144 |

Training of MAE-MoCo and MAE-Nextclip take approximately 77 hours and 42 hours, respectively. Additionally, while MAE-MoCo has 27M parameters, the number of trainable parameters in MAE-Nextclip is 15M.

*2) Fine-tuning:* The pretrained models, using all three datasets mentioned in Section II-B1, are fine-tuned on Chapman dataset. After pretraining, the encoders of the models are retained, while the remaining components (momentum

encoder and decoder for MAE-MoCo, two decoders for MAE-Nextclip) are discarded. It should also be noted that the *projection head* of the MAE-MoCo encoder is not utilized for fine-tuning purposes.

A linear network with a sufficient number of output nodes capable of classifying the appropriate dataset is connected to the `[CLS]` tokens of the pretrained encoders. The resulting architecture is tuned using two distinct techniques, namely the **linear probe** and **fine-tuning** methods. In the **linear probe** approach, all trainable parameters of the pretrained encoder are frozen, and only the parameters of the new linear network connected to the `[CLS]` token are updated during task-specific training. Consequently, in this mode, the success of pretraining is more accurately assessed since the encoder functions as a feature extractor, while the linear layer acts as a linear classifier. Conversely, in the **fine-tuning** mode, the encoder is initialized with pretrained parameters, but both the encoder and linear layer parameters are trained end-to-end during task-specific training.

The objective of the models in fine-tuning is to minimize the cross-entropy loss between the one-hot encoded ground truth category labels and the probabilities outputted by the model. In fine-tuning, as in pretraining, warm-up epochs and learning rate schedulers are utilized for similar purposes. Additionally, early stopping is employed during fine-tuning. If no improvement in validation loss is observed over a certain number of epochs, the fine-tuning training is terminated. The model with the best performance in terms of validation loss is selected and used for testing. The detailed configuration of the fine-tuning is provided in Table VII

TABLE VII
FINE-TUNING CONFIGURATION

| Batch size | 64 |
|---|---|
| Optimizer | Adam |
| Learning rate | $1 * 10^{-3}$ |
| Warm-up epochs | 20 |
| Early stopping epochs | 15 |
| Learning rate schedular: reduce rate | 0.9 |
| Learning rate schedular: number of bad epoch | 5 |

*C. Definition of the Metrics*

Accuracy is a metric commonly encountered in nearly all classification tasks, shedding light on the model's classification ability. Accuracy is calculated by the ratio of correctly predicted individual samples to the total number of test samples [26].

It is noteworthy that the calculation of accuracy does not account for any imbalance in the data distribution among the classes in the test dataset. Thus, macro-F1 score and area under curve of receiver operating characteristic (AUC) come into play. The macro-F1 score is computed as the harmonic mean of precision and recall [26]. Precision measures the classifier's ability to not misclassify negative examples as positive, while recall represents the ability to find all positive examples [26].

AUC, on the other hand, is a metric that demonstrates the classifier's performance at different thresholds and aims to measure how well positive and negative examples are separated. AUC is a metric defined for binary classifiers; hence, in this study, since the classifiers are multi-label classifiers, one-vs-one AUC is employed. One-vs-one AUC computes the AUC metric by averaging over all possible class pairs.

## III. RESULTS

The proposed models need to be compared with previous and state-of-the-art self-supervised learning methods in terms of metrics to demonstrate the effectiveness of the models. Metrics commonly used in model comparison, such as accuracy, macro-F1 score, and AUC, are utilized in comparison. The MAE-MoCo and MAE-Nextclip models fine-tuned with Chapman dataset are initially compared with the state-of-the-art on published benchmark that are specific to the field of ECG. Subsequently, MAE-MoCo and MAE-Nextclip are compared with previous prominent self-supervised learning methods such as MTAE [19], Nextclip [20], MoCo [15], and SimCLR [14].

*A. Comparison with State-of-the-art on Published Benchmarks*

MAE-MoCo and MAE-Nextclip are compared with Mass-MIB [27], a leading method in generative self-supervised learning demonstrating state-of-the-art performance on Chapman dataset, and ISL [17], representing the contrastive learning aspect in the field of ECG. Comparison results on the Chapman dataset is presented in Tables VIII. MassMIB and ISL utilize the higher-level classes of the Chapman dataset; hence, the comparisons are conducted with 4 classes, and the AUC metric is included. The configuration of Chapman dataset with 4 classes is called as Chapman-Reduced for clearance.

TABLE VIII
COMPARISON OF THE MAE-MOCO AND MAE-NEXTCLIP WITH
STATE-OF-THE-ART SELF-SUPERVISED LEARNING FRAMEWORKS ON
CHAPMAN-REDUCED DATASET

| Frameworks | Linear Probe | | Fine-tuning | |
|---|---|---|---|---|
| | *Macro-F1* | *AUC* | *Macro-F1* | *AUC* |
| ISL w/o Inter [17] | N/A | 0.764±0.011 | N/A | 0.989±0.001 |
| ISL w/o Intra [17] | N/A | 0.921±0.030 | N/A | 0.989±0.003 |
| ISL [17] | N/A | 0.965±0.008 | N/A | 0.991±0.001 |
| MassMIB [27] | N/A | N/A | 0.950±0.000 | N/A |
| MAE-Nextclip ($\alpha = 1$) | **0.967±0.000** | **0.995±0.000** | **0.972±0.001** | **0.998±0.000** |
| MAE-MoCo ($\alpha = 40$) | 0.961±0.003 | 0.978±0.003 | 0.968±0.003 | **0.998±0.000** |

When comparing the results, it is observed that in the Chapman dataset, the proposed models have outperformed both MassMIB framework using fine-tuning technique in terms of macro-F1 by 0.02 and ISL models in terms of AUC by 0.03 in linear probing and by 0.007 points in fine-tuning. MassMIB employs masking in both time and frequency domains to discover generative features, while ISL employs intra-patient and inter-patient discrimination to achieve contrastive features. In proposed models, the extraction of both contrastive and generative features is the most significant factor in outperforming ISL and MassMIB.

Furthermore, it is observed that the MAE-Nextclip model performs better than MAE-MoCo in the Chapman dataset.

One possible reason behind this is that one of the most important features of the Nextclip model is its successful detection of atrial fibrillation. Since a significant portion of the Chapman dataset consists of atrial fibrillation, MAE-Nextclip outperforms MAE-MoCo in terms of metrics.

### B. Methodological Comparison with Other Self-supervised Learning Strategies

MAE-MoCo and MAE-Nextclip demonstrated robust and effective performance when compared to state-of-the-art on published benchmarks, surpassing the compared methods on Chapman dataset. Moreover, the metric comparisons of MAE-MoCo and MAE-Nextclip models that are pretrained with CPSC2018, PTB-XL, and Chapman datasets, with MTAE [19], Nextclip [20], MoCo [15], and SimCLR [14] utilizing linear probe and fine-tuning techniques on Chapman dataset, are presented in Table IX. The compared models are implemented from scratch with the configuration used by MAE-Nextclip and MAE-MoCo for a fair comparison. It should be noted that in this comparison, unlike some literature like MassMIB and ISL where classes are merged into a higher-level class to reduce the number of classes to 4, the original Chapman dataset is utilized without merging the labels. Due to the large number of classes in the Chapman dataset, AUC is not used for comparison.

TABLE IX
COMPARISON OF THE MAE-MoCo AND MAE-NEXTCLIP WITH
PREVIOUS SELF-SUPERVISED LEARNING FRAMEWORKS ON CHAPMAN
DATASET

| Frameworks | Linear Probe | | Fine-tuning | |
|---|---|---|---|---|
| | *Accuracy* | *Macro-F1* | *Accuracy* | *Macro-F1* |
| SimCLR [14] | 0.707±0.002 | 0.508±0.001 | 0.847±0.011 | 0.695±0.009 |
| MoCo [15] | 0.767±0.008 | 0.612±0.004 | 0.848±0.004 | 0.716±0.021 |
| Nextclip [20] | 0.900±0.002 | 0.751±0.021 | 0.952±0.003 | 0.860±0.001 |
| MTAE [19] | 0.947±0.001 | **0.851±0.008** | **0.958±0.002** | 0.870±0.024 |
| MAE-Nextclip ($\alpha = 2$) | 0.948±0.000 | 0.844±0.023 | 0.956±0.002 | 0.862±0.009 |
| MAE-Nextclip ($\alpha = 1$) | **0.949±0.001** | **0.848±0.029** | 0.955±0.002 | 0.863±0.007 |
| MAE-Nextclip ($\alpha = 0.5$) | 0.943±0.000 | 0.830±0.013 | 0.952±0.002 | 0.858±0.019 |
| MAE-MoCo ($\alpha = 40$) | 0.943±0.003 | 0.848±0.006 | 0.955±0.002 | 0.862±0.015 |
| MAE-MoCo ($\alpha = 20$) | 0.937±0.002 | 0.844±0.004 | 0.955±0.004 | 0.871±0.012 |
| MAE-MoCo ($\alpha = 10$) | 0.944±0.004 | 0.835±0.013 | **0.957±0.001** | **0.888±0.002** |

Upon examination of Table IX, it is evident that MAE-MoCo and MAE-Nextclip outperform previous methods. The proposed models exhibit a noticeably superior performance across all metrics compared to other self-supervised learning methods. This is primarily attributed to the optimization of two different tasks during pretraining. Although the resulting models from pretraining are capable of performing two tasks rather than excelling in a single task, the individual tasks operate with slightly lower performance. The proposed models achieve a balanced performance between these tasks. Therefore, the learned representations are less task-specific and more universal, thereby preserving their potential for fine-tuning. Consequently, when these pretrained encoders are used in fine-tuning, they approach the performance of optimal classifiers more closely.

For different $\alpha$ values, it is seen that the MAE-MoCo performs better, especially with lower $\alpha$ values, due to its effective regularization against overfitting in fine-tuning mode,

particularly on the smaller dataset such as Chapman dataset. For MAE-Nextclip, the model improves performance with $\alpha = 1$ due to stronger masking features, as opposed to moving overfitting region when $\alpha$ increased from 0.5 to 1.

## IV. DISCUSSION

### A. Limitations

The MAE-MoCo and MAE-Nextclip frameworks are indeed powerful models that outperform other models, but they have some limitations such as high computational costs and the challenge of finding optimal parameter sets.

Due to the fact that MAE-MoCo and MAE-Nextclip frameworks consist of a combination of two different pretext tasks, the computational cost required for training is higher compared to individual pretext tasks. Specifically, contrastive models have even higher costs compared to generative models due to the double forward pass. Consequently, MAE-MoCo and MAE-Nextclip can be considered more costly than other models.

Another limitation of the proposed frameworks is the increase in the number of non-trainable parameters resulting from the combination of two different models. The increase in the number of non-trainable parameters complicates the task of finding the optimal parameter set compared to other individual models.

### B. Future Work

To further enhance the performance of the proposed frameworks, future work requires the utilization of different, more advanced backbones, the combination of different tasks, the utilization of different types of data to test models and more hyperparameter tuning.

The convolutional vision transformer (CvT), incorporating convolutions into the ViT architecture [28], has proven its efficacy on datasets like ImageNet and different backbones like CvT, ResNet could serve as a promising candidate for performing combined tasks more effectively in the ECG domain.

Additionally, the combined tasks can be altered or enhanced, with alternative tasks such as Centerclip which reconstructs randomly selected segments of ECG data rather than the next segment, SimCLR, BYOL, DINO [29]. Moreover, another strategy to improve the performance of combined self-supervised learning tasks could be increasing the number of tasks.

Furthermore, experimenting the models with other physiological signals that have ECG signal characteristics such as being a time series and a 2D image can better demonstrate the effect of the combination strategy.

Finally, more optimal fixed parameters can be found with more computational power since MAE-MoCo and MAE-Nextclip have many fixed parameters. Although the values used are the most optimal ones in the original frameworks, they may not be the most ideal values for MAE-MoCo and MAE-Nextclip, so the proposed models may have more optimized versions.

## V. Conclusion

In this paper, novel pretraining tasks are introduced from the perspective of self-supervised learning, arising from the significant problems of obtaining labeled data. The MAE-MoCo framework is proposed to combine the strengths of generative self-supervised learning, exemplified by MAE, and contrastive learning, as seen in MoCo, thus allowing for the simultaneous utilization of generative and contrastive learning. Additionally, MAE-Nextclip emerges from the merge of two distinct generative learning tasks, MAE and Nextclip. The fundamental factor contributing to the emergence of both models is the [CLS] token within ViT which serves as an aggregate representation. While the MAE task inherently operates at the level of individual patch tokens to reconstruct masked patches, the utilization of information from the [CLS] token suffices for Nextclip and MoCo tasks.

The fine-tuning results demonstrate that the proposed models are achieving new state-of-the-art results on the Chapman dataset. The primary reason for outperforming other models lies in the concurrent extraction of generative and contrastive features with the assistance of the pretrained encoder in the proposed models.

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
