# OpenReview forum: "Combined self-supervised learning for ECG classification"
_IEEE.org/EMBS/BHI/2024/Conference — IEEE BHI'24_

### Official Review · Reviewer_fyb4 · 2024-08-08
**A technically solid paper, but might need further evaluation and clarification**

**Overall Rating:** 7
**Confidence:** 4

**Other Quality Metrics:**

(a) Clarity of writing - good;
(b) Clinical Significance - good;
(c) Methodological Novelty - great;
(d) Experiments and Results - good

**Questions For The Authors:**

- How is the computational complexity of the proposed method? How is the pretraining time compared to other SSL methods such as SimCLR or BYOL?
- I might missed it somewhere, but what is the performances of purely supervised method? The tables only covers the benchmarks from SSL methods.

**Strengths:**

- This paper introduces a novel approach that combines generative and contrastive self-supervised learning methods for ECG classification. The proposed MAE-MoCo and MAE-Nextclip frameworks demonstrate superior performance compared to state-of-the-art self-supervised learning methods on ECGs with the Chapman dataset, which indicates the effectiveness of the combined approach.
- This paper is well-written and organized, with clear explanations of the proposed methods, experimental setup, and results.

**Summary Of The Paper:**

This paper proposes a novel approach to address the challenge of limited labeled data in ECG classification by combining generative and contrastive self-supervised learning methods. The authors introduce two variants of the framework, MAE-MoCo and MAE-Nextclip, which leverage the Vision Transformer (ViT) as backbones. MAE-MoCo combines a masked autoencoder with a momentum encoder for generative and contrastive learning, while MAE-Nextclip combines two generative learning tasks, masked autoencoding and Nextclip reconstruction. The models are pre-trained on a large unlabeled ECG dataset and then fine-tuned on a smaller labeled dataset for classification. The results demonstrate that the proposed models outperform state-of-the-art self-supervised learning methods on the Chapman dataset, highlighting the potential of combining pre-training tasks for improved ECG signal representation and classification.

**Weaknesses:**

- The evaluation is primarily conducted on the Chapman dataset. Further validation on other publicly available ECG datasets would strengthen the generalizability of the findings.
- Lack some necessary ablations. The choice backbones might need an comprehensive ablation for reasoning. As indicated in some previous ECG-based deep learning studies, the convolutional architecture might still outperform the ViTs in processing ECG data. I am wondering how will the performances be if the the authors use CNN structures, such as ResNet, as the backbone encoder.

---

### Official Review · Reviewer_nhwF · 2024-08-10
**ECG self-supervised learning for classification, MAE-MoCo and MAE-Nextclip**

**Overall Rating:** 5
**Confidence:** 4

**Other Quality Metrics:**

Clarity of Writing: Fair
Clinical Significance: Fair
Methodological Novelty: Good
Experiments and Results: Fair

**Questions For The Authors:**

1. Clarity of Title and Core Concepts: The term "combined self-supervised learning" in the title is unclear (what exactly is being combined): more specifically, the core idea of leveraging generative+contrastive SSL, as well as using the [CLS] token as an aggregate representation. Could you clarify and better convey these points in the title and body of the paper? Impact: Clarifying this with some additional text could enhance the paper's readability and ensure the core concepts are well-understood.
2. Could you provide more details on the computational resources required for training the proposed models?How sensitive are the models to changes in hyperparameters (especially the parameters for Transformer architecture)? This would help assess the feasibility for other researchers. Impact: Clarifying the model size and/or FLOPS could influence the evaluation of the model's practicality and accessibility.
3. Hyperparameter Sensitivity: How sensitive are the models to changes in hyperparameters? Have you performed any sensitivity analysis? Impact: Understanding hyperparameter sensitivity would provide insights into the robustness of the models and their ease of use.

**Strengths:**

* Innovative combination of methods: The integration of generative and contrastive learning within the same framework is novel and shows promise in improving representation learning for ECG classification.
* Strong Performance: The proposed models outperform existing state-of-the-art methods on the Chapman dataset, with significant improvements in macro-F1 score and AUC. The models are pretrained on (a combination of) diverse ECG datasets, which enhances the reliability of the results.

**Summary Of The Paper:**

This paper introduces two novel self-supervised learning frameworks, MAE-MoCo and MAE-Nextclip, for ECG classification. The frameworks leverage a Vision Transformer (ViT) architecture and incorporate the [CLS] token to improve representation learning. MAE-MoCo combines generative and contrastive learning using a masked autoencoder (MAE) and a momentum encoder (MoCo). MAE-Nextclip enhances generative self-supervised learning by reconstructing both masked patches and Nextclip data. The methods are validated on a joint database, including CPSC2018, PTB-XL, and Chapman datasets.

**Weaknesses:**

1. Core concept communication and unclear title: The title "Combined self-supervised learning for ECG classification" is somewhat vague. It is not immediately clear what methods are being combined. The core idea of using the [CLS] token as an aggregate representation should be better emphasized and clearly conveyed in both the title and the body of the paper.
2. High computational cost, and the complexity in hyperparameter tuning: The combination of multiple pretext tasks increases the computational burden, making the models less accessible for researchers with limited resources. The models (with Transformer backbones) have a large number of hyperparameters, which may require extensive tuning to achieve optimal performance. This complexity could be a barrier to reproduction and application in other settings.
3. (minor issue) Generalization beyond ECG data: The applicability of the proposed methods to other types of physiological signals or medical data is not explored, which could limit the impact of the work.

---

### Official Review · Reviewer_4s6x · 2024-08-12
**The paper proposes the idea of combined self-supervised learning for ECG classification**

**Overall Rating:** 7
**Confidence:** 4

**Other Quality Metrics:**

(a) Clarity of writing : good
(b) Clinical significance: fair
(c) Methodological Novelty: good
(d) Experimental and Results:  good

**Questions For The Authors:**

1. The authors are requested to justify the source of equation (9), (10),(11) and (12).

**Strengths:**

1.The paper is well-written,
2. The paper is well-conceptualized
3. The paper  well-structured
4. The paper is well-supported by experimental results and justifications.

**Summary Of The Paper:**

The paper proposes the idea of constrictive and generative self-supervised learning methods such as vision learning transformer (ViT) neural network architecture to resist the dependency on expensive labeled data for cardiovascular disease classification.

**Weaknesses:**

1. The Table VIII. and IX seems bit confusing and saturated, The authors are requested to enhance the quality of visibility and make these more clear.

---

### Decision · Program_Chairs · 2024-09-23

Accept